# Approximating the Performance of a Time-Domain Pulsed Induction EMI Sensor with Multiple Frequency-Domain FEM Simulations for Improved Modelling of Arctic Sea-Ice Thickness

**DOI:** 10.3390/s25237317

**Published:** 2025-12-01

**Authors:** Becan Lawless, Danny Hills, Adam D. Fletcher, Liam A. Marsh

**Affiliations:** Department of Electrical and Electronic Engineering, University of Manchester, Manchester M13 9PL, UK; danny.hills@manchester.ac.uk (D.H.); adam.fletcher@manchester.ac.uk (A.D.F.)

**Keywords:** finite element modelling, FEM, electromagnetic sensing, time-domain metal detection, optimisation, sensor simulation, sensor characterisation

## Abstract

One of the key challenges with developing pulsed induction (PI) electromagnetic induction (EMI) sensors for use in the Arctic is the inaccessibility of the environment, which makes in situ testing prohibitively expensive. To mitigate this, sensor development can be streamlined through the creation of a robust simulation strategy with which to optimize features such as coil turns and geometry. Building on work that previously presented a method for simulating an Arctic PI sensor via a time-domain finite element model (FEM), this paper presents a method for approximating a time-domain simulation with multiple frequency-domain simulations. A comparison between the fast Fourier transform (FFT) of a time-domain simulation and a collection of frequency-domain simulations is presented. These are validated against empirical data with a PI sensor over seawater, with an air gap used as a proxy for sea ice. Using the method described, a range of coils is simulated with dimensions from 0.5×0.5 m up to 1.0×2.0 m, demonstrating the ability of this approach to enable comparison of sensor performance over a wider parameter space. For a parametric sweep over 10 sensor-to-seawater lift-off distances, the improvement from the time-domain simulation (of a 402 μs window) to the frequency-domain simulation (comprising 100 discrete frequencies) represents a reduction in simulation time from 38,013 min to 141 min.

## 1. Introduction

Sea-ice thickness (SIT) can be measured in several ways. The most accurate method is by direct measurement of destructively drilled holes, bored through the ice with an auger [1]. This approach relies on skilled human operators and therefore suffers from a lack of scalability for recording large amounts of data or data spanning vast geographic regions. Satellite-based measurements can be used to cover large areas; however, this approach relies on several assumptions, including hydrostatic equilibrium, snow-layer coverage, and sea-ice density [2]. In addition, localised SIT measurements need to provide calibration and validation data to support satellite-based measurement techniques. Local data collection offers a higher resolution of measurements within a specific region of interest and allows better mapping of subsurface terrain compared to satellite measurements.

One of the most common methods of taking localised, non-destructive measurements of sea-ice thickness is to use electromagnetic induction (EMI) sensors [3]. The use of EMI exploits the conductivity of seawater and the contrast with sea ice (which is electrically non-conductive at the frequencies utilised by inductive sensors) [4]. The EMI sensor is sensitive to the resultant magnetic field due to eddy currents induced in the seawater by a primary excitation field. The magnitude of the received signal decreases as the distance between the sensor and the water surface increases, and so the magnitude of the received signal can be calibrated to give an estimate of the thickness of the ice that separates the sensor from the seawater [5].

EMI sensors are used in a wide range of applications for which the methodology described could be applicable, including security, non-destructive testing, surveying, archaeology, and material characterisation [6,7,8,9]. EMI sensors can be configured to operate in either the frequency domain using “continuous-wave” excitation or the time domain using pulsed excitation.

It is possible to implement either a time-domain or a frequency-domain sensor for SIT measurement. Each approach has associated advantages and limitations. For example, frequency-domain systems can take advantage of resonance to improve signal-to-noise ratio but have an inherently limited bandwidth as a result. Time-domain systems have far greater bandwidth to interrogate the surrounding area, but sometimes at a cost to sensitivity due to the transmit pulse energy being spread across a wide range of frequencies. In this application, there is a limited number of available commercial EMI-based sensors. One such sensor, the Geonics EM31, has been extensively used to measure SIT [10,11]. This system uses multiple coils acting in a transmit–receive arrangement and with a single fixed frequency of 9.8 kHz [12]. As pulsed induction systems are inherently broadband, they enable the isolation of frequency components that are more adversely affected by vibration [13,14]. Furthermore, the transient nature of pulsed induction allows easier implementation of single-coil systems; a single coil removes the possibility of coil-to-coil positional changes as a result of vibration. However, modelling a coil with such a mixed role presents some complexities, which are considered as part of this study. One study implemented a system that was both towable and sled-mounted. In a towed or sled-mounted system, and in such a challenging natural environment, vibration is anticipated to be a significant concern [15].

Due to the high cost and logistical challenges of deploying hardware to remote polar environments, there is a clear need for robust simulation techniques to aid development. Finite element modelling (FEM)-based simulations provide the capability to construct complex geometries, and so are preferred for this type of application. Previous work [16] has developed a method to overcome the issue of using a single coil to both apply and measure a magnetic field in the time domain. This approach uses a two-stage FEM-based simulation using COMSOL Multiphysics. The ‘excitation study’ is run first, modelling the full-field inductive effects over a body of saltwater large enough to be assumed infinite. The ‘response study’ then uses the induced currents simulated in the seawater to produce the secondary-field effects, which give rise to the secondary voltage in the sensing coil. It is necessary to approach this with multiple studies since COMSOL Multiphysics does not inherently distinguish between primary and secondary-field effects [17].

Although simulating the entire time-domain response is useful for understanding the specific response of the EMI system to the seawater present under the ice, it requires a large number of time steps to properly resolve the transmit pulse. For the system described in this study, a 402 μs window is needed to resolve the representative PI system response resulting from the 200 μs pulse shown in Figure 1. This is a smoothed version of the raw current profile measured via shunt presented in Figure 2. Even using high-performance computers, this simulation can take several days to complete a single simulation. This can extend to weeks of computational time where there is a need to generate data that can illustrate trends, such as how the magnitude of the received signal decays as the sensor is moved further from the seawater. This becomes prohibitive when performing multi-iteration parametric analyses to optimise coil characteristics.

Time-domain simulations become computationally expensive due to the number of time steps needed to accurately discretize the window under consideration. When modelling a PI system, the rapid change in the transmit current pulse necessitates a large number of steps to properly resolve the pulse edges. In general, frequency-domain simulations are less computationally expensive since they are limited to a specific band of frequencies. This naturally reduces the number of constituent models required; however, the bandwidth of the simulation is reduced to only the harmonics simulated.

In [18], the authors used a frequency-domain simulation approach in COMSOL Multiphysics to solve a Landau–Lifshitz–Gilbert equation for magnetic dynamics instead of the more traditional time-domain approach. In order to compare these approaches, they ran time- and frequency-domain simulations of spin-wave dispersions for identical models. Afterward, a Fourier transform was used to extract the frequency spectrum from the simulated time-domain data. The frequency-domain simulation was around four times faster and “more accurate than the time-domain simulation”. In simulations with relatively large time windows, combined with significant transient behaviour requiring a short simulation time step, the simulation cost between time-domain and frequency-domain simulations can grow significantly.

It is therefore desirable to have a method for comparing the sensitivity of coils using frequency-domain simulations as an approximation of time-domain scenarios, which will be outlined in this paper. This paper sets out to establish that a frequency-domain approach can result in a reduced computation time, enabling the generation of larger datasets and thereby providing a more robust basis for comparison of sensor performance. This also allows for simulations to become more geometrically complex and better reflect real-world scenarios, which could include features such as brine inclusions in the ice layer or non-parallel arrangements of the sensor and water surface. A reduction in simulation time also represents a significant saving in energy consumption.

## 2. Modelling and Simulation

### 2.1. Generating Current Profiles

In [16], it was shown that in order to undertake a COMSOL single-coil PI simulation and establish the secondary-field component of the coil voltage, the simulation can be split into a full-field study and a secondary-field study. This requires the input of measured current data for each coil geometry. For this purpose, three coils were constructed from a light foam board, 15 turns of 0.52 mm^2^ PVC-insulated wire, and a wooden frame. The coil dimensions were 0.5×1.0 m, 0.5×1.5 m, and 1.0×1.0 m, and are illustrated in Figure 3.

Each coil was wired in series with a shunt resistor of 1 Ω to allow for current measurement. The current measurement data series contained 32,768 data points over a window of 299 μs. To minimise noise, 256 consecutive waveforms were averaged. This resulted in a raw current profile for each coil, three of which are shown in Figure 2.

To create the full-field time-domain study in COMSOL, a time-varying current input is required. The raw current profile contains a large amount of very high-frequency noise; including this in the simulation requires very small time steps to converge. Simulating noise that is beyond the bandwidth of the experimental system results in high computational cost. To facilitate this, a smoothing function was applied to the current profile, which reduced computational time. The current profiles were first downsampled to 200 points. A piecewise cubic interpolation function was then fit to generate the smoothed data, resulting in the three current profiles shown in Figure 1. A comparison of the FFT from each of the two datasets is presented in Figure 4, which demonstrates that the smoothing function had only a minimal impact on the magnitudes of the frequencies represented within the waveforms in the inductive response band. At 336 kHz, the magnitude of the smoothed current FFT output dropped to around $10^{-18}$ A due to smoothing removing these components. This occurred before resonance at approximately 430 kHz. The percentage error between the FFT outputs from the pre- and post-smoothing currents increased from around 0.01% at 1 kHz to 0.9% at 50 kHz and up to 3.4% at 100 kHz. These errors are percentages of significantly reduced magnitudes, so their effect on the average magnitude was minimal.

### 2.2. Time-Domain Simulation

As outlined in [16], a time-domain simulation takes an empirically gathered current profile as input and applies it to a coil above a body of seawater. Two models of the coil are used to isolate the secondary voltage arising from the currents induced in the seawater. In the excitation study, the coil is driven by the empirically determined (and smoothed) current pulse. In the response study, the coil is defined as an open circuit. The simulation first calculates the current distribution in the seawater using the first model, then isolates the secondary voltage response received by the coil by selecting the open-circuit model and allowing the calculated current distribution to decay.

An example of the current density in the seawater is shown in Figure 5. The system is modelled with an air gap approximating the presence of sea ice, since in most cases, the salinity of sea ice is such that its conductivity is negligible compared to seawater [19]. The equivalent representation of the underlying mesh can be seen in Figure 6.

The Magnetic Fields solver within the COMSOL AC/DC module implements Maxwell’s equations. These are extensively documented in [20]. The simulation is defined with magnetic insulation boundary conditions.

Since the sensor consists of a rectangular coil over an essentially infinite body of seawater, the two planes of symmetry can be utilised through appropriate boundary plane conditions, thereby representing the full 3D simulation via a quarter model [21]. The depth of the water and the radius of the cylinder representing the water and air are set such that at the outer ’magnetic insulation’ boundaries, the induced electrical current is negligible. Having set the model size, the mesh is adjusted to have a higher density in the region of the highest-fidelity requirement: the seawater region directly below the coil.

Utilising the key parameters outlined in Table 1, a time-domain simulation was performed for seawater-to-coil lift-off distances of 0.6 m to 2.4 m. This range approximately matches common thickness ranges of first-year Arctic sea ice [22]. Using the formulas described in [16], the external current density values were input into the seawater domains of the response study from the excitation study. The time intervals for the outputs were set to every $1\times 10^{-7}$ s. All simulations were conducted in COMSOL version 6.3, using a computer with 128GB of RAM and an NVIDIA (NVIDIA, Santa Clara, CA, USA) Quadro RTX6000 graphics card.

### 2.3. Frequency-Domain Simulation

The simulation for the frequency-domain model is in most respects identical to the time-domain variant, with a few key differences. Instead of requiring a pointer for time-dependent results to be selected from the excitation study, a frequency-dependent result is required. The volume of seawater below the coil is defined to have an external current density, which is determined from a prior simulation when the coil is driven by an external current of 1 A. The frequency-domain magnitudes are rescaled post-simulation based on the empirically measured current.

A final key difference is that instead of a time-domain study, a frequency-domain study is set for a range of frequencies. In this case, the chosen range was from 1 kHz to 100 kHz in 1 kHz increments. This is in keeping with the range identified in [14] to be the region of key sensitivity. Simulating the 10 lift-off separations from 0.6 m to 2.4 m at 0.2 m increments in the time domain took 38,013 min. Achieving the same in the frequency domain for 100 distinct frequencies took 141 min on the same hardware. This represents an approximately 99.6% reduction in simulation time and a similar reduction in energy cost. This ratio is significantly larger than the one presented in [18]. This is in part because, in order to simulate approximately the full PI time-domain response, a window of 2.401 ms is required. Further, while the output time step is $1\times 10^{-7}$ s, the actual COMSOL simulation update time step is even smaller, generally around $1.25\times 10^{-8}$ s, with a maximum of $2.5\times 10^{-8}$ s. This value is set dynamically such that the residual errors arising from transients are sufficiently low, and it is decreased automatically by the software as needed. These simulation time steps represent a Nyquist frequency of 40 MHz, and although not all of this information is stored, these short simulation time steps represent a significant increase in the computational time. By performing an FFT on the time-domain simulation outputs, as shown in Figure 7, it can be seen that the clearest separation between the responses occurred at frequencies less than ≈300 kHz. This clearly shows that a simulation with time steps of $1.25\times 10^{-8}$ s, and containing frequencies up to 40 MHz, contains a large amount of frequency data that has a magnitude much less than that required to distinguish the effects of lift-off.

In the simulation, each frequency is driven by the same magnitude current. This differs from the experimental coil, where the current pulse splits energy over a wide bandwidth. In order to relate the simulated frequency-domain results to the actual coil performance, each frequency magnitude must be scaled proportionally to that component of the measured current. To generate these scaling factors, an FFT was performed on the current profiles shown in Figure 2, thus identifying the components of the input current corresponding to each frequency in the range. Since the raw current profile contained 32,768 values with a time increment of $9.15525\times 10^{-9}$ s, the initial frequency resolution would be around 3.3 kHz. To bring this in line with the 1 kHz spacing of the simulation, the current profile was zero-padded to 109,227 data points, bringing the total time to 1 ms and the frequency resolution to 1 kHz. This generated a dataset derived directly from the true EMI sensor parameters, thereby providing a mechanism for comparison with measured data.

In order to compare the simulated frequency-domain and time-domain datasets, an FFT was performed on the output of the time-domain simulation. To remain consistent with the previous method, the time-domain secondary voltage values were also zero-padded to 10,000 data points, which, for a time step of $1\times 10^{-7}$ s, represents an output frequency resolution of 1 kHz.

To compare the two sets of scaled frequency-domain magnitudes as a function of lift-off, a mean was taken of the values, giving a single frequency-domain secondary voltage magnitude for each lift-off. The values for the simulated frequency-domain data and the FFT output of the simulated time-domain data are presented in linear and log plots in Figure 8 and Figure 9, respectively.

The results show excellent correlation between the outputs of the time-domain and frequency-domain simulations. The same qualitative trends can be seen to agree for all three geometries in Figure 8. The logarithmic plot in Figure 9 provides the most insight, showing that agreement holds across the full range of lift-off separations, with clear separation between coil geometries. Calculations of the root mean square error (RMSE) indicate that the error between the time- and frequency-domain simulations can be quantified to a worst case of 12.4 μV and a mean value of 0.84% of the simulated secondary voltage. These differences are considered negligible compared with the magnitude of the corresponding secondary voltage values, which is between x and y.

### 2.4. Frequency-Domain Simulation Harmonic Sensitivity

This work has shown that for this use case, frequency-domain simulation is significantly faster than time-domain simulation for establishing the relationship between the secondary voltage and lift-off seawater for a given system.

The work discussed previously employed 100 frequencies in the range of 1 kHz to 100 kHz. This frequency resolution was chosen in order to balance computational cost with the need to capture the features of the FFT plot, such as that in Figure 4. In order to assess whether the frequency resolution could be further reduced, a comparison was made by reducing the frequency resolution to 50, 25, 10, and 5 data points. Figure 10 illustrates the results of this study. The separation is minor, which becomes clearer when looking at Figure 11, which shows the average RMSE between the 100 data-point set and the subsequent thinned datasets. This shows that there is little additional RMSE (relative to 100 data points) introduced by thinning the data to 50 data points, indicating that a sufficient sample density has been achieved. Below this, the RMSE starts to increase. This is likely because the multiplicative scale factor now starts to overfit the lower-frequency components to compensate for the change in overall magnitude. Therefore, the increase in RMSE below 50 sample points indicates that this is not an accurate representation of the time-domain signal.

## 3. Results and Discussion

### 3.1. Experimental Validation

Utilising the same procedure as outlined in [14], data was gathered for a new electrical design and the three foam, wire, and timber coil geometries described in Figure 3. This data was recorded to enable comparison of the two methods of simulation to measured data and to provide a baseline for the sensitivity limit of the coils. Lift-off measurements were taken from 1.2 m above the waterline up to 2.4 m. The minimum value was selected to ensure that the sensors were free from saturation during measurement. This experiment involved the construction of a completely metal-free, floating pontoon structure able to raise and lower assorted EMI sensors to known heights above a pool of seawater. Experiments were conducted over a man-made tidal pool, which naturally refills with seawater when the tide is sufficiently high, and was used in order to maintain a calm water surface and avoid tidal depth changes. Lift-off was measured from the centreline of the coil construction to the waterline using weighted fabric tape measures, with an estimated uncertainty of ±5 mm. The seawater in the pool was measured to have a conductivity of 64.3 mS/cm, approximately the same as that of the nearby sea at 62.0 mS/cm. The experimental arrangement is shown in Figure 12.

The experimental data has a window of 1024 data points over 204.6 μs with a time step of 0.2 μs. This would produce an FFT output with a frequency resolution of 4882 Hz. By zero-padding to 5000 data points, or a 1 ms window size, this resolution is increased to 1 kHz. The padding can be performed to this extent due to the transient nature of the signal, meaning the vast majority of its frequency components are seen in the initial measured window. This padding improves the accuracy of frequency magnitudes as more discrete frequency bins are resolved. Padding data in this way causes a transition in the time-domain waveform where the padding starts, appearing as injected high frequencies in an FFT. To reduce the effect of this, frequency bins above 100 kHz in the FFT output are ignored for processing. Frequency bins below 4882 Hz are also not used for processing, as these correspond to wavelengths larger than the sample window. In order to remove any response not caused by seawater lift-off changes, the measurement with the lowest-magnitude response was set as the reference value for each dataset through subtraction. This was the 3.0 m lift-off for both the 0.5×1.0 m and 1.0×1.0 m coils; however, for the 0.5×1.5 m coil, this was found to be the 2.8 m measurement. This is believed to be due to issues with the pontoon frame holding the longer coil steady at 3.0 m. This applied stress caused distortion of the coil, which was seen as a false increase in the response signal. For the purposes of comparison, these minimum values represent an approximation of the profile of the signal under free-space conditions. In order to compare the resultant trends with the simulated data, each of the measured datasets was multiplied by a scale factor, set such that the mean value of each dataset was equal. This process accounts for gain stages and digitisation in the experimental system through a single multiplicative calibration. The resultant comparisons between the simulated frequency domain and measured values can be seen in Figure 13, Figure 14 and Figure 15.

When plotting the results for the larger coils, it was found that there was a sudden divergence at lower lift-off values, with the simulated responses appearing much higher in magnitude than the measured responses. This was thought to be due to amplifier saturation at these lift-off values, meaning the response signal was clipped and appeared smaller than in reality. This effect is discussed further in [14]. Because of the divergence due to amplifier clipping, values below 1.2 m for the 0.5×1.5 m and 1.0×1.0 m coils and below 1.0 m for the 0.5×1.0 m coil were not plotted to keep the region of interest as large as possible. The simulated and experimental measurements showed a good trend correlation when processed as described. Notably, the trend was less clear for the 1.0×1.0 m coil; this may be due to the large area of this coil, which made it more difficult to keep the panel steady during measurements. At lower lift-off values on the 0.5×1.5 m and 0.5×1.0 m coils, the measurements showed some small differences compared to the simulation. This was thought to be a result of the aforementioned amplifier saturation at lower lift-off distances still having some effect. Some deviations potentially arose from limitations in the simulation, which was inherently idealised. The simulation assumed the seawater to be approximately infinite in all directions, perfectly flat, and in a fully metal-free environment. The simulation also assumed no inclination or movement of the coil, which proved challenging to achieve with actual experimental measurements.

### 3.2. Comparison of Simulation-Only Coils

The primary use case for the method described in this paper is the timely comparison of different coil geometries or EMI sensing scenarios through simulation alone. Currently, the method requires the experimental measurement of the current profile through a shunt; however, this can be achieved in ways other than building a permanent physical analogue. In order to create temporary coil geometries, a pegged board was constructed with removable pegs, with holes spaced at 100 mm to a maximum size of 1.1×2.3 m, as shown in Figure 16. To obtain a representative sample of geometries to showcase the method described in this paper, the current profiles of different-sized coils were measured. Each coil comprised 15 turns of a 0.52 mm^2^ cable, as listed in Table 2.

It is difficult to create a single unified noise floor for a wide range of coil geometries because each has a geometry-dependent level of sensitivity to background noise. However, when the coils are of similar dimensions (as in this work), and the standard deviation of each coil’s measurements is similar, an approximate noise floor can be determined. This allows comparisons to be made between different coils. A noise floor for the system can then be defined as the point at which a measurement’s absolute magnitude falls within three standard deviations of zero; any measurement above this threshold can be considered the response of a measurand rather than random noise present in all measurements. By combining the datasets’ standard deviations, it was found that the three-standard-deviation noise floor was 37.4 μV. In practice, this is a conservative estimate of noise, as the focus on comparing datasets has been prioritised over methods of noise mitigation. The noise floor of the electronic hardware is the minimum achievable value of the system noise floor, which is orders of magnitude lower than the stated noise floor of 37.4 μV. Therefore, there remain several options to lower the stated noise value using methods such as filtering and advancements in data post-processing.

Utilising the same method as described above, the mean of the frequency magnitudes (scaled using the shunt current-derived scaling factors) was compared for seven unique coil geometries. The trends for these coils with lift-off from the body of seawater are presented in Figure 17 up to a height of 5 m, along with the standard-deviation-derived theoretical measurement noise floor. The lift-off value at which the respective coil curves intersect the noise floor is presented in Table 3, represented as black crosses in Figure 17. These serve as a demonstration of the method and its ability to compare relative coil performance.

## 4. Conclusions and Future Work

The results presented in this paper demonstrate that it is possible to represent a time-domain system as a collection of frequency-domain simulations with extensive time and energy savings and minimal additional error. This approach was validated by comparing the lift-off trends of equivalent configurations in the time and frequency domains, plus an additional comparison to measured data from a proxy scenario to SIT measurement. The representation of time-domain simulation through a series of frequency-domain simulations is of particular benefit for its ability to significantly increase the rate at which different coil geometries, parameters, and scenarios can be compared in the context of their influence on SIT. Simulation of a coil first requires a representative coil to be constructed and its current profile experimentally measured via a shunt resistor. The measured current can be used to scale the frequency-domain simulation outputs. Zero-padding is used to ensure that the frequency resolution is maintained between the time and frequency domains. Due to the time window being relatively large in comparison with the step required to capture the EMI transients, a change from the time domain to the frequency domain represents a 99.6% reduction in simulation time in this scenario. This is potentially an extreme example, but it clearly demonstrates the significant benefit of this approach in the right context.

By looking at the noise floor of a given electronics system, the decay curve for voltage versus lift-off from seawater can be used to estimate a theoretical limit to the sensitivity of the measurement system when paired with a specific coil. This has potential benefits for situations where manufacturing costs are prohibitive or where access to representative testing conditions is limited. With further work, it may be possible to remove the need to experimentally verify current profiles for arbitrary coil geometries prior to conducting FEM simulations. A potential method may be to develop SPICE models that can synthesise this data based on pre-defined parameters such as the coil dimensions, number of turns, and electrical properties of the conductor.

The method in this paper could allow for the inclusion of other important sea-ice characteristics within the FE model. This could play a key role in improving estimates of SIT measurement errors by quantifying the impact of a range of uncertainties. Current EMI sensors for SIT measurement have no way to adapt to uncertainties such as sensor inclination or brine inclusions. Currently, data gathered with an inclined EMI sensor is affected at an unknown rate. For this reason, measurements are largely gathered on flat areas of ice in the hopes that the incline of the system remains sufficiently low for measurements to be accurate. An EMI SIT measurement system, when combined with accelerometer data and a model for the effects of inclination on secondary voltages in the coil, could, in principle, allow for a wider range of valid data-gathering locations and provide an indication of when data reaches a sufficiently high uncertainty to be unusable.

Future work will involve the application of the described method to additional SIT measurement scenarios and comparisons with data collected under Arctic conditions.

## Figures and Tables

**Figure 1 sensors-25-07317-f001:**
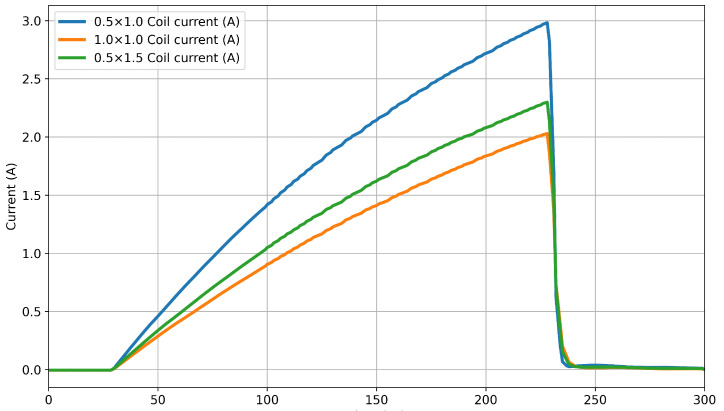
Measured coil excitation current after smoothing, used as input for time-domain simulation via an interpolation function.

**Figure 2 sensors-25-07317-f002:**
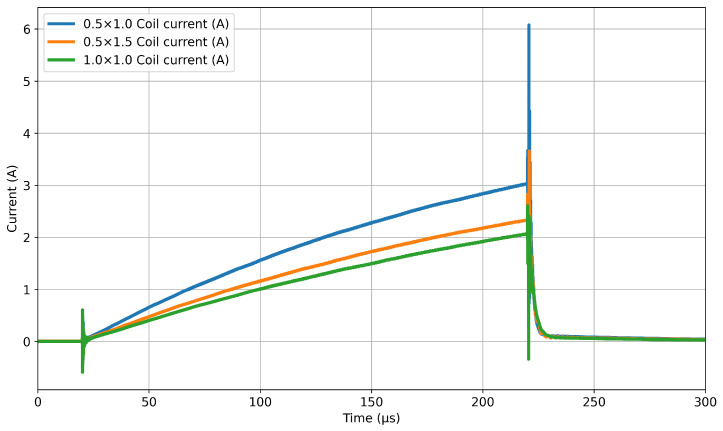
Measured coil excitation current before smoothing, used as input for the FFT to create weightings for frequency-domain simulation.

**Figure 3 sensors-25-07317-f003:**
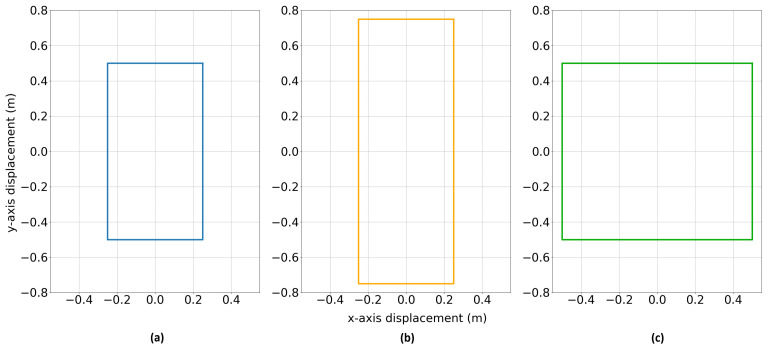
Coil geometries used to collect current profiles through a shunt resistor and in field research over an ocean-water tidal pool: (**a**) 0.5 × 1.0 m, (**b**) 0.5 × 1.5 m, (**c**) 1.0 × 1.0 m.

**Figure 4 sensors-25-07317-f004:**
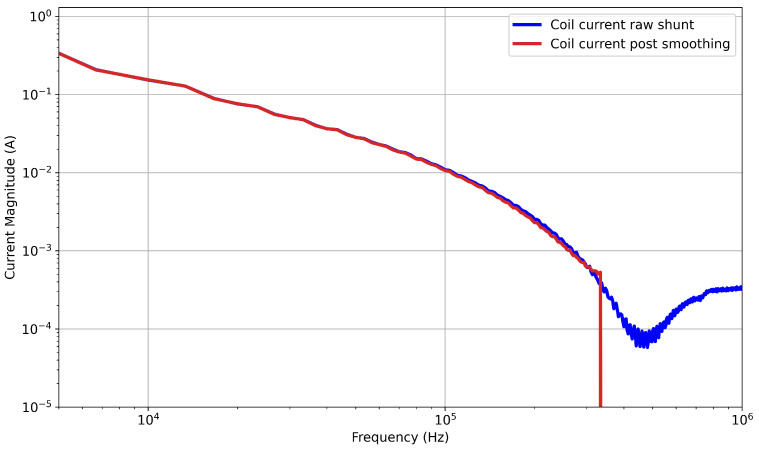
Comparison of FFTs from current data presented in Figure 1 and Figure 2 for a 0.5×1.0 m coil.

**Figure 5 sensors-25-07317-f005:**
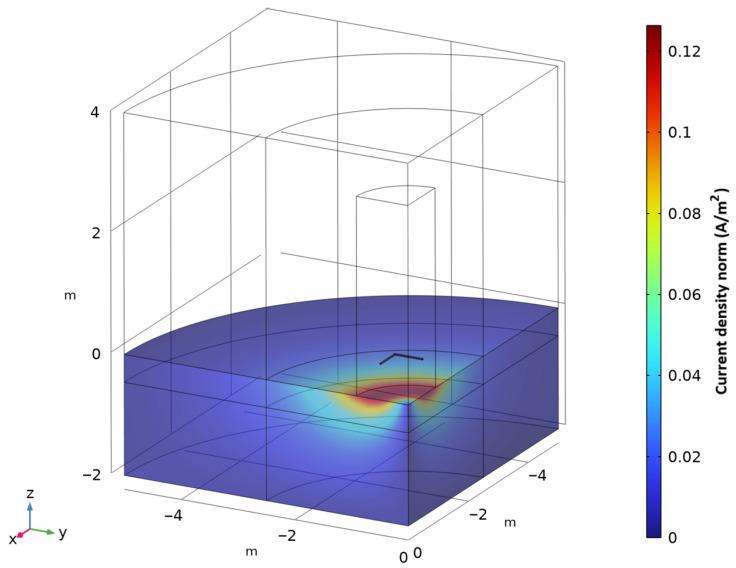
COMSOL example of the current density induced in seawater domains to be used as inputs to the second study.

**Figure 6 sensors-25-07317-f006:**
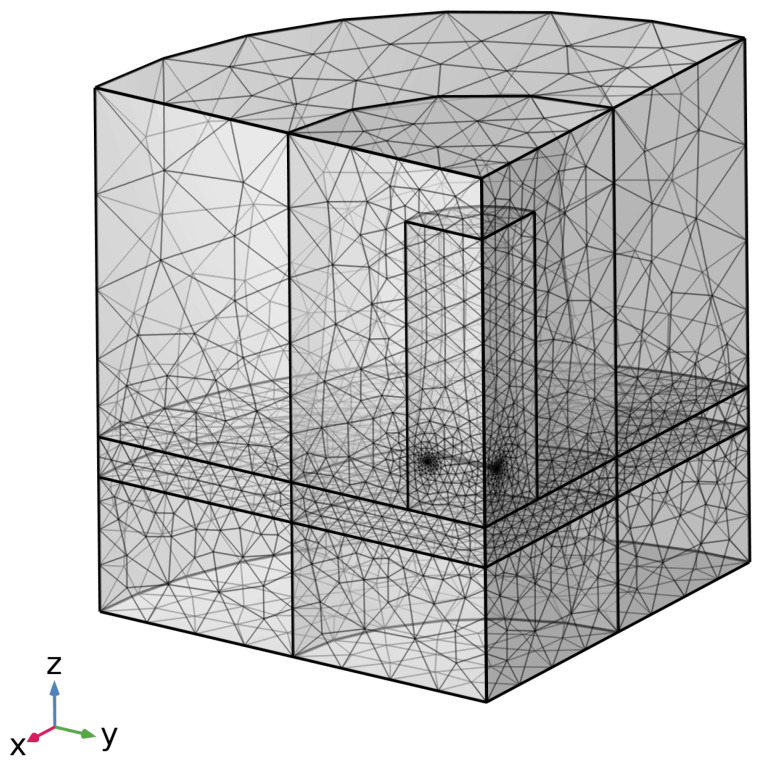
COMSOL example of mesh distribution for a 0.6 m lift-off.

**Figure 7 sensors-25-07317-f007:**
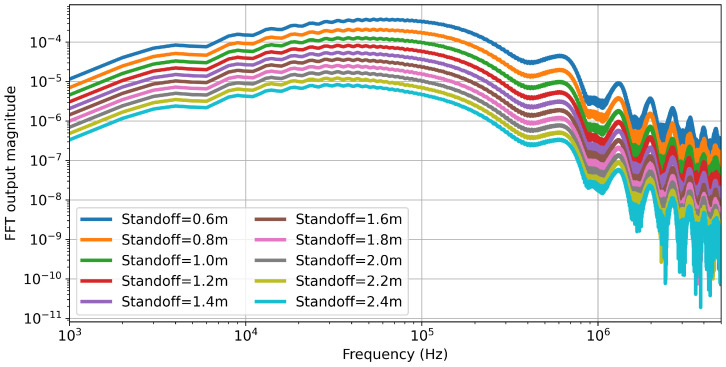
FFT of time-domain simulation secondary voltage output for the 0.5×1.0 m coil.

**Figure 8 sensors-25-07317-f008:**
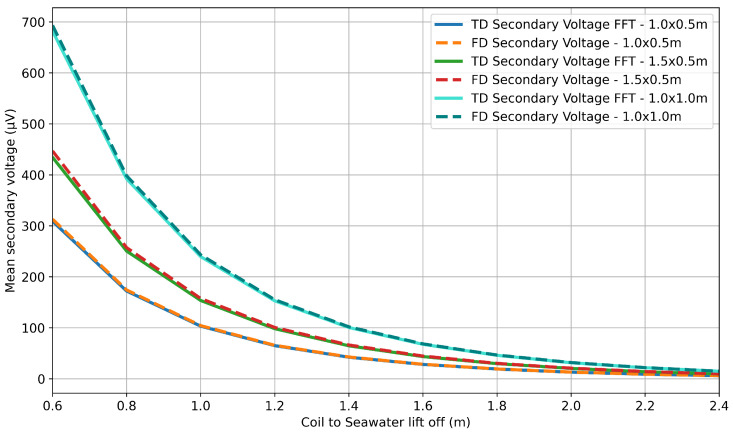
Comparison of time-domain FFT and frequency-domain simulation data: linear plot.

**Figure 9 sensors-25-07317-f009:**
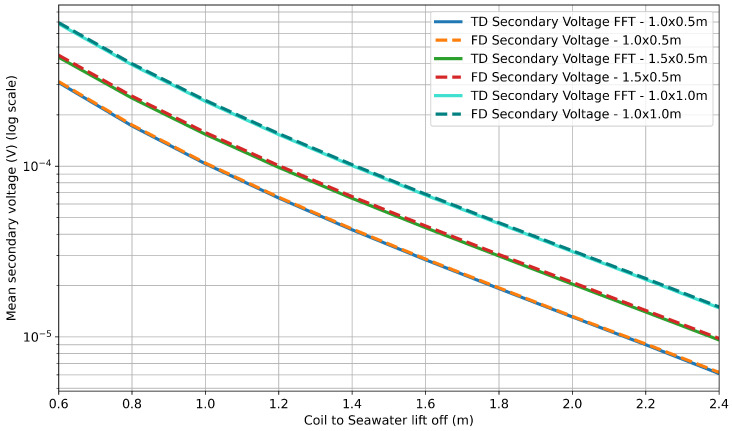
Comparison of time-domain FFT and frequency-domain simulation data: logarithmic plot.

**Figure 10 sensors-25-07317-f010:**
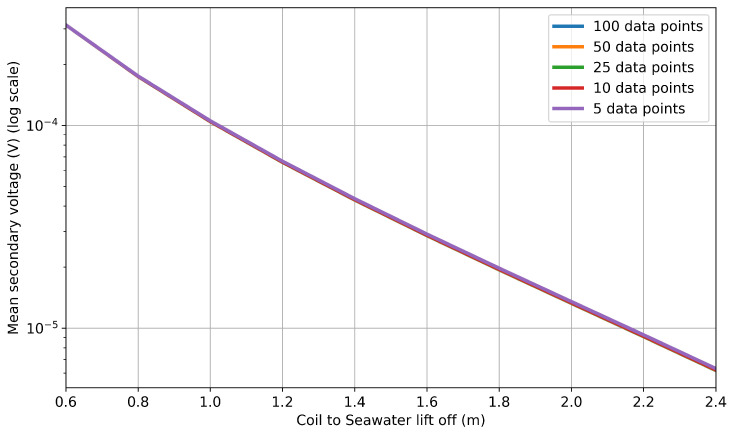
Comparison of lift-off above seawater versus voltage relations for the 0.5 × 1.0 m coil with reduced frequency resolution between 1 and 100 kHz.

**Figure 11 sensors-25-07317-f011:**
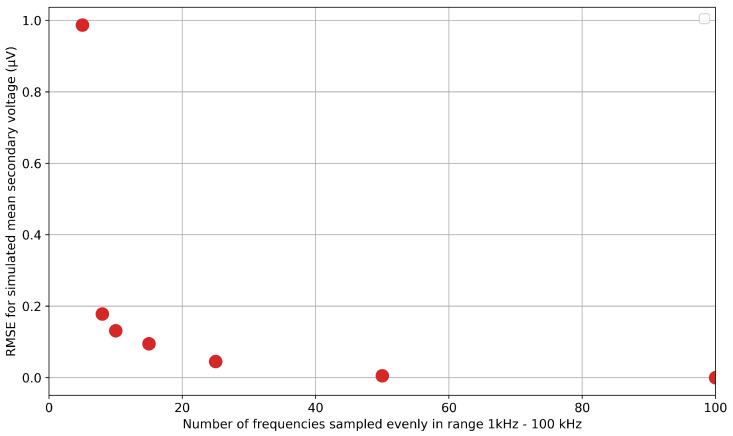
Average RMSE, normalized to the minimum secondary voltage, against the number of frequencies sampled in the range of 1–100 kHz.

**Figure 12 sensors-25-07317-f012:**
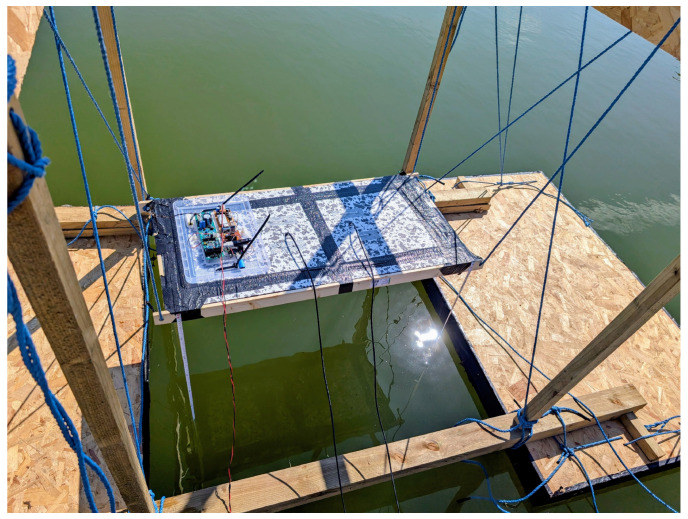
Experimental rig consisting of a metal-free floating pontoon with which assorted EMI sensors could be raised and lowered over seawater in a tidal pool.

**Figure 13 sensors-25-07317-f013:**
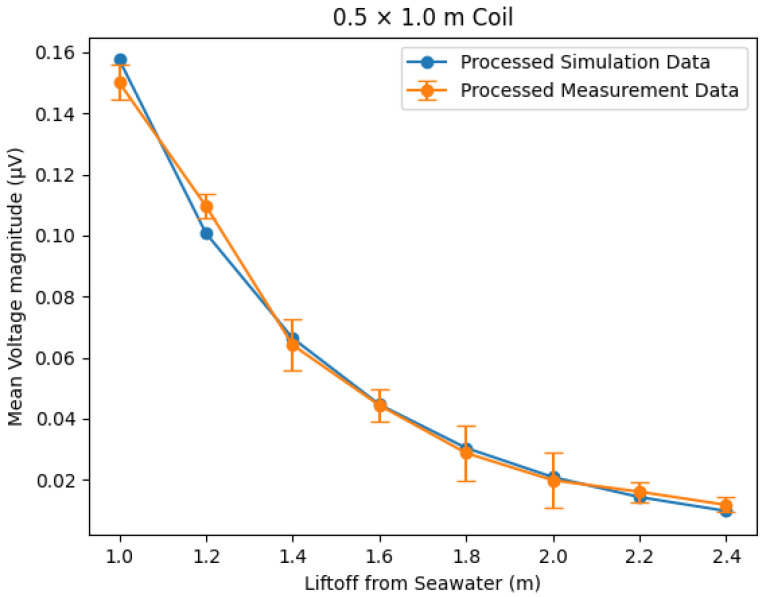
Trend comparison of lift-off and secondary voltages for the 0.5 × 1.0 m coil between the simulated frequency-domain response and the measured response. Error bars = ±3 standard deviations.

**Figure 14 sensors-25-07317-f014:**
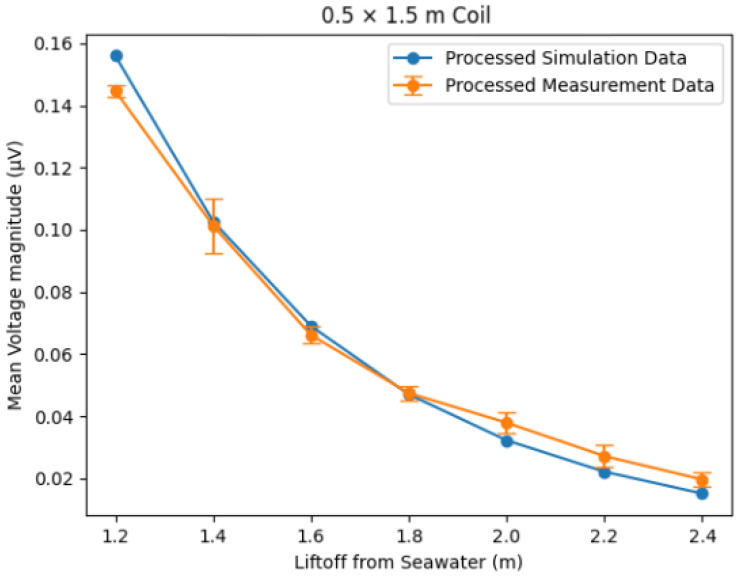
Trend comparison of lift-off and secondary voltages for the 0.5 × 1.5 m coil between the simulated frequency-domain response and the measured response. Error bars = ±3 standard deviations.

**Figure 15 sensors-25-07317-f015:**
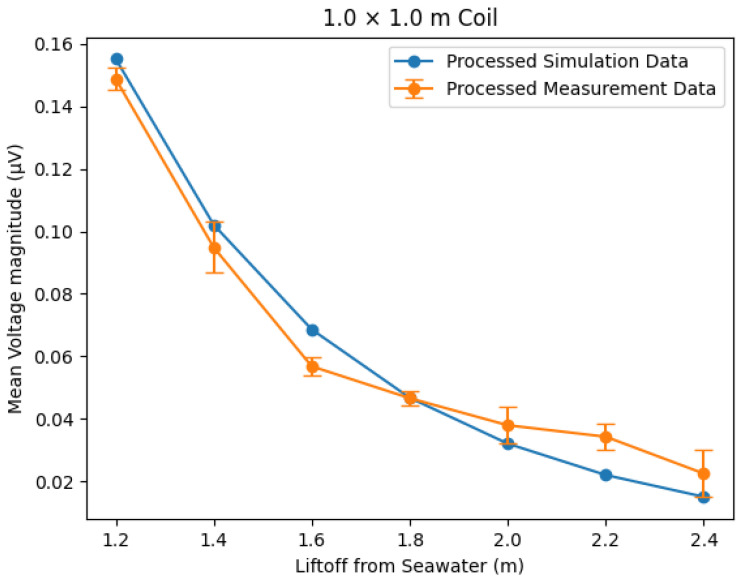
Trend comparison between lift-off and secondary voltages for the 1.0 × 1.0 m coil between the simulated frequency-domain response and the measured response. Error bars = ±3 standard deviations.

**Figure 16 sensors-25-07317-f016:**
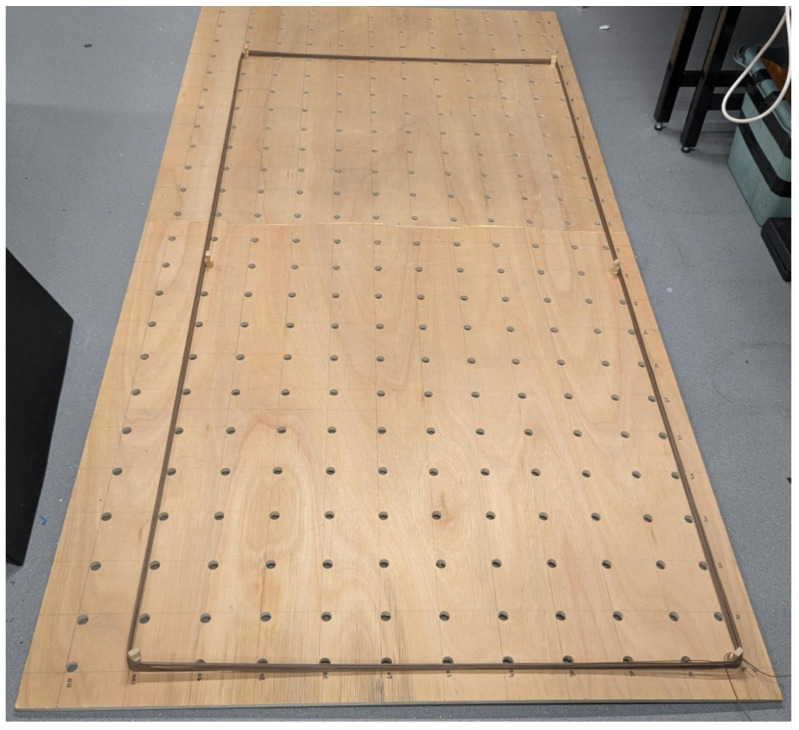
Pegged board used to build a temporary 1.0 × 2.0 m, 15-turn coil for gathering current data via a shunt.

**Figure 17 sensors-25-07317-f017:**
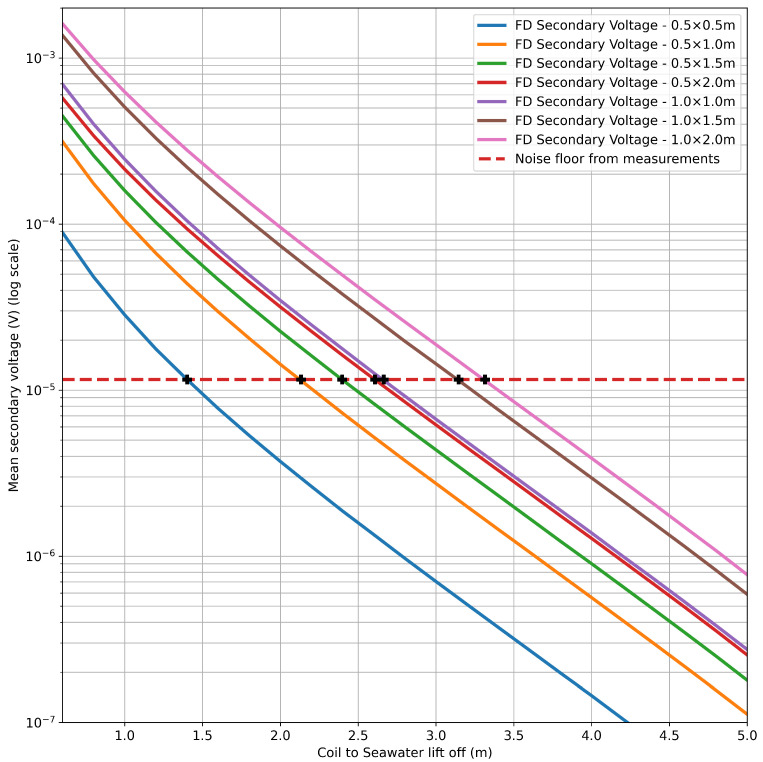
Comparison of all coil simulations in the frequency domain to the noise floor: logarithmic plot.

**Table 1 sensors-25-07317-t001:** Key parameters for COMSOL 3D time-domain simulation.

Description	Value	Units
Seawater electrical conductivity	6.43	S/m
Seawater relative permeability	1	
Seawater relative permittivity	80	
Multiturn coil number of turns	15	
Wire cross-sectional area	$5.2\times 10^{-7}$	m^2^
Solver	MUMPS	
Time-dependent solver maximum timestep	$2.5\times 10^{-8}$	s
Time-dependent study user-defined relative tolerance	0.01	

**Table 2 sensors-25-07317-t002:** Geometries of current profiles sampled.

0.5 × 0.5 m	0.5 × 1.0 m	0.5 × 1.5 m	0.5 × 2.0 m
1.0 × 0.5 m	1.0 × 1.0 m	1.0 × 1.5 m	1.0 × 2.0 m

**Table 3 sensors-25-07317-t003:** Theoretical sensitivity limits for simulated coil geometries based on the noise floor (8.15 μV).

Coil Geometry (m)	Approximate Lift-Off at Noise Floor (m)
0.5 × 0.5	1.40
0.5 × 1.0	2.13
0.5 × 1.5	2.40
0.5 × 2.0	2.61
1.0 × 1.0	2.66
1.0 × 1.5	3.15
1.0 × 2.0	3.31

## Data Availability

The original data presented in the study is openly available via DOI https://doi.org/10.48420/30500387, accessed on 25 November 2025, under the licence CC BY-NY 4.0.

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
