# Peer review of "Approximating the Performance of a Time-Domain Pulsed Induction EMI Sensor with Multiple Frequency-Domain FEM Simulations for Improved Modelling of Arctic Sea-Ice Thickness"

_sensors, 2025, doi:10.3390/s25237317_

Round 1

Reviewer 1 Report

Comments and Suggestions for Authors

The manuscript presents a valuable approach for accelerating the simulation of pulsed induction electromagnetic induction (PI-EMI) sensors. To further strengthen the work, the following points should be carefully addressed:
1. The weighting process uses the magnitude of the current's FFT. However, the time-domain response is dependent on both the amplitude and phase of the frequency components. Does using only the magnitude for weighting introduce any potential error? Was the phase response of the system considered or analyzed? 

2. The experimental validation uses an air gap to proxy sea ice. Real sea ice, however, may contain brine inclusions with non-negligible conductivity. How might the presence of such inclusions affect the accuracy of the proposed simulation method?

3. Figure 5 compares the FFT before and after smoothing, which is very helpful. Which frequency components were mainly filtered out by the smoothing process? Could it be possible to briefly indicate the significantly suppressed noise bands in Figure 5 in the form of illustrations or annotations, so that readers can understand the effect of smoothing more intuitively?

4. When conducting frequency domain simulation, why is the frequency range set to 1 - 100 kHz? Why not consider higher or lower frequencies? Please explain your reasons.

When conducting frequency domain simulation, in which the frequency range is set to 1 - 100 kHz. While in Figure 7 the frequency is out of this range.

5. For the proposed EMI sensor, why is a rectangular coil used instead of a circular one or any other shape? Is the simulation method proposed in this article applicable to coils of other shapes?

Author Response

The manuscript presents a valuable approach for accelerating the simulation of pulsed induction electromagnetic induction (PI-EMI) sensors. To further strengthen the work, the following points should be carefully addressed:
1. The weighting process uses the magnitude of the current's FFT. However, the time-domain response is dependent on both the amplitude and phase of the frequency components. Does using only the magnitude for weighting introduce any potential error? Was the phase response of the system considered or analyzed? 

In our time domain simulations we use a measured time-varying current as the input which inherently incorporates magnitude and phase. This is therefore fully incorporated in the time domain simulations. For the frequency domain simulation we only use the magnitude.

When combining the signals we only consider the magnitude in both instances which in the frequency domain is independent of phase and so does not introduce additional error.

When analysing the measurements there is minimal appreciable phase shift with liftoff. Any phase shift we see due to liftoff appears within the noise floor. That might change with future iterations as we start to incorporate information about additional complications such as brine inclusions however that is outside of the scope for this study.

  1. The experimental validation uses an air gap to proxy sea ice. Real sea ice, however, may contain brine inclusions with non-negligible conductivity. How might the presence of such inclusions affect the accuracy of the proposed simulation method?

The comparison in this paper considers an idealised case of the fully non-conductive sea ice. Consequently, in this paper the comparison is a simulated air gap to measured air gap. It is certainly true that brine inclusions will have some effect on the inductive measurements of an EM sea ice thickness sensor. In multi-year ice the presence of brine is minimal and so less likely to cause detrimental effect, but with first year ice the effect is thus far uncharacterised. That will form future work for this group, however any simulation of a complex system relies on verification of the more simple variations first. It is not foreseen that the inclusion of additional conductive brine elements will reduce the comparability of equivalent time domain and scaled frequency domain simulations.

  1. Figure 5 compares the FFT before and after smoothing, which is very helpful. Which frequency components were mainly filtered out by the smoothing process? Could it be possible to briefly indicate the significantly suppressed noise bands in Figure 5 in the form of illustrations or annotations, so that readers can understand the effect of smoothing more intuitively?

The high frequency elements in Figure 3, which have been effectively filtered out through smoothing in Figure 4 are spread across a large spectrum of frequencies, though low in magnitude. The window for Figure 5 has been increased to show more clearly the effect smoothing has on the higher frequencies. The following text has been added for additional context of the error values, clearly showing noise suppression above 330kHz appearing as a brick wall filter. Additional text has been added to lines 133 to 136.

  1. When conducting frequency domain simulation, why is the frequency range set to 1 - 100 kHz? Why not consider higher or lower frequencies? Please explain your reasons.

When conducting frequency domain simulation, in which the frequency range is set to 1 - 100 kHz. While in Figure 7 the frequency is out of this range.

On lines 193 – 196 in the paper it is stated that the separation is clearest for values below 300kHz. Figure 8 aims to illustrate the need for the banding out frequencies above 300kHz. As for why 1 – 100kHz were chosen specifically, this was largely because of the work in (Hills 2025) which showed that the inductive region is between 500 Hz and 200 kHz, and resonance occurs around 350 kHz. Since the response of seawater is largely uniform with frequency this band was tightened somewhat to reduce signal to noise.

  1. For the proposed EMI sensor, why is a rectangular coil used instead of a circular one or any other shape? Is the simulation method proposed in this article applicable to coils of other shapes?

Yes, the process should be generally applicable with appropriate model. Our coil is rectangular because it maximises the area which could fit within the footprint of a sled, and is mechanically easier to construct. A rectangle is also good because it's a simple case which is not axially symmetric so presents a slightly more non-trivial modelling problem.

Reviewer 2 Report

Comments and Suggestions for Authors

Non-destructive testing sensors are an important tool for assessing the condition of study objects. This paper addresses an interesting issue related to recording ice thickness in the Arctic. This is a crucial aspect for assessing climate change and analyzing the impact of conditions, geography, geology, and external factors on the state of the surrounding system. Modeling is one of the main methods in this paper. Modeling the behavior of electromagnetic coils, taking into account the time factor, leads to significant computational and time expenditures. A frequency analysis approach allows for faster assessment. Field studies were also conducted. The results are new and will be of interest to the scientific community.

Comments and recommendations:

  1. The introduction should clearly outline the purpose and objectives of the work, as well as outline the research plan.
  2. The calculation scheme should be provided, specifying the initial and boundary conditions. The mathematical formulation should also be included.
  3. The finite element partitioning should be characterized. The computing equipment used for the calculations should be identified, including its capacity. Was the influence of the grid on the solution assessed?
  4. Parameter value spikes at times of approximately 20 and 220 are observed in Figure 3. Explain these effects.
  5. Parameter fluctuations in Figure 7 require explanation.
  6. A critical analysis of the results is missing; a discussion section is required. Modeling limitations should also be noted.

Author Response

Reviewer 2

Non-destructive testing sensors are an important tool for assessing the condition of study objects. This paper addresses an interesting issue related to recording ice thickness in the Arctic. This is a crucial aspect for assessing climate change and analyzing the impact of conditions, geography, geology, and external factors on the state of the surrounding system. Modeling is one of the main methods in this paper. Modeling the behavior of electromagnetic coils, taking into account the time factor, leads to significant computational and time expenditures. A frequency analysis approach allows for faster assessment. Field studies were also conducted. The results are new and will be of interest to the scientific community.

Comments and recommendations:

  1. The introduction should clearly outline the purpose and objectives of the work, as well as outline the research plan.

We feel that the introduction provides sufficient context for the purpose of this research, which can be inferred from the descriptions in lines 82 – 88 where the benefit of frequency domain is established. Additional text has been added on lines 101 to 104, in order to highlight the section of the introduction which outlines the research plan, and specifically state the purpose of the work.

  1. The calculation scheme should be provided, specifying the initial and boundary conditions. The mathematical formulation should also be included.

The equations from the main physics node and boundary conditions have been added via a citation of the relevant COMSOL documentation. This outlines the implementation of Maxwell’s equations in the AC/DC module of COMSOL.

  1. The finite element partitioning should be characterized. The computing equipment used for the calculations should be identified, including its capacity. Was the influence of the grid on the solution assessed?

A representation of the mesh used for a 0.6m lift off has been added as Figure 7 to provide additional illustration of the FE partitioning. A statement about the computing equipment specifications has been added on line 167. “All simulations were conducted in COMSOL version 6.3, using a computer with 128GB ram, and a NVIDIA Quadro RTX6000 graphics card.”

  1. Parameter value spikes at times of approximately 20 and 220 are observed in Figure 3. Explain these effects.

These artefacts are common high frequency, low magnitude ringing resulting from trying to push a large di/dt through an inductive load. They are a standard artefact of pulsed induction systems.

  1. Parameter fluctuations in Figure 7 require explanation.

Minor fluctuations in the data between 10 and 100 kHz is likely a result of FFTing a sampled representation of the TD signal. Spectral leakage can occur when there are sharp edges in a time domain signal, manifesting as ripples in the frequency domain signal.

  1. A critical analysis of the results is missing; a discussion section is required. Modeling limitations should also be noted

Section 3 presents a combined results and discussion section which we feel best structures the presentation of the results in this paper. Discussion of the role that the limitations of simulation might play in the deviation between simulated and measured results has been added on lines 301 to 305.

Reviewer 3 Report

Comments and Suggestions for Authors

The reviewed article addresses a current topic in the field of electromagnetic nondestructive evaluation, or a specific subgroup of this field. The authors present an innovative approach to numerical modelling, focusing on approximating the performance of a TD pulsed induction EMI sensor through multiple frequency-domain FEM simulations. This unique approach is a significant contribution to the field, aiming to evaluate and analyse ice thickness. 

After studying the article, I conclude that the contribution is self-consistent.

 I have the following specific comments on the article to enhance its professionalism and appeal. These comments are minor and can be easily implemented at the revision level. They do not detract from the overall quality of the work. 

1: Table 1: when stating the dimensions of relative quantities /relative permittivity and conductivity/, “ul” is indicated. Despite the understandable meaning, I recommend using [-] because the symbolism used can be slightly confusing. The same applies to the parameter “Multiturn coil number of turns”. At the same time, I would like to ask the authors to modify this name: the word “multiturn” seems vague. An inductor with more than one turn is a “multiturn” inductor. I leave it to the authors’ discretion.
2. Figure 1: From a formal and technical point of view, the figure lacks units of quantities. The figure gives a very “cheap” impression. 

3. Figure 11: The adjacent functional values ​​in the figure are connected by a straight line, indicating a linear dependence. The text accompanying the figure does not explain anywhere why this is done this way. It would be more appropriate to use linear regression or similar methods, because the assumption of a straight line is only an approximation. As an example of how the graph should look, figures 14 and 15 show the correct display using a box plot. The authors should explain the significance of the linear dependence and why it was chosen over other methods. 

4. It is appropriate to provide details in the article on how the excitation signals for the EMI sensor were defined in the COMSOL program. Given the nature of the excitation signal in Figure 1b, it is assumed that the first derivative at the falling edge of the signal will be discontinuous. Please add to the text a clear explanation of the nature of the excitation signal and how it was defined in the COMSOL program. 

All comments raised are of a nature that does not dramatically reduce the expertise of the article. Therefore, the authors should consider incorporating them. After incorporating the comments, I recommend the article for publication.

Author Response

Reviewer 3

The reviewed article addresses a current topic in the field of electromagnetic nondestructive evaluation, or a specific subgroup of this field. The authors present an innovative approach to numerical modelling, focusing on approximating the performance of a TD pulsed induction EMI sensor through multiple frequency-domain FEM simulations. This unique approach is a significant contribution to the field, aiming to evaluate and analyse ice thickness. 

After studying the article, I conclude that the contribution is self-consistent.

 I have the following specific comments on the article to enhance its professionalism and appeal. These comments are minor and can be easily implemented at the revision level. They do not detract from the overall quality of the work. 

1: Table 1: when stating the dimensions of relative quantities /relative permittivity and conductivity/, “ul” is indicated. Despite the understandable meaning, I recommend using [-] because the symbolism used can be slightly confusing. The same applies to the parameter “Multiturn coil number of turns”. At the same time, I would like to ask the authors to modify this name: the word “multiturn” seems vague. An inductor with more than one turn is a “multiturn” inductor. I leave it to the authors’ discretion.

I have removed “UL”  from units in the described table as suggested.
2. Figure 1: From a formal and technical point of view, the figure lacks units of quantities. The figure gives a very “cheap” impression. 

I have removed this figure and changed the text referring to it.

  1. Figure 11: The adjacent functional values ​​in the figure are connected by a straight line, indicating a linear dependence. The text accompanying the figure does not explain anywhere why this is done this way. It would be more appropriate to use linear regression or similar methods, because the assumption of a straight line is only an approximation. As an example of how the graph should look, figures 14 and 15 show the correct display using a box plot. The authors should explain the significance of the linear dependence and why it was chosen over other methods. 

The figure has had the lines removed and 2 additional data points added. Hopefully this demonstrates more clearly that while a trend is apparent, we aren’t specifying exactly what the value of that trend is.

  1. It is appropriate to provide details in the article on how the excitation signals for the EMI sensor were defined in the COMSOL program. Given the nature of the excitation signal in Figure 1b, it is assumed that the first derivative at the falling edge of the signal will be discontinuous. Please add to the text a clear explanation of the nature of the excitation signal and how it was defined in the COMSOL program. 

This is referred to on line 127 in that the down-sampled current measurements are supplied to COMSOL which then applies a “piecewise cubic interpolation function” through these data points. Some additional context has been added on the mathematical models used in the form of a reference to the relevant COMSOL documentation.

All comments raised are of a nature that does not dramatically reduce the expertise of the article. Therefore, the authors should consider incorporating them. After incorporating the comments, I recommend the article for publication.